# Relationship between Choroidal Thickness and Anterior Scleral Thickness in Patients with Keratoconus

**DOI:** 10.3390/diagnostics14202280

**Published:** 2024-10-14

**Authors:** Neus Burguera-Giménez, M.ª Amparo Díez-Ajenjo, Celeste Briceno-Lopez, Noemí Burguera, M.ª José Luque-Cobija, Cristina Peris-Martínez

**Affiliations:** 1Anterior Segment, Cornea and External Eye Diseases Unit, Fundación de Oftalmología Médica (FOM), Av. Pío Baroja, 12, E-46015 Valencia, Spain; amparo.diez@uv.es (M.A.D.-A.); celeste.briceno@uv.es (C.B.-L.); cristina.peris@fom.es (C.P.-M.); 2Department of Optics, Optometry and Vision Sciences, Physics School, University of Valencia, Dr. Moliner, 50, E-46100 Valencia, Spain; maria.j.luque@uv.es; 3Ophthalmology Department, Q Vision, Vithas Virgen del Mar Hospital, Ctra. el Mami a Viator, Km.1, E-04120 Almería, Spain; noemiburgueraid@qvision.es; 4Surgery Department, Ophthalmology, School of Medicine, University of Valencia, Av. Blasco Ibáñez, 15, E-46010 Valencia, Spain

**Keywords:** choroidal thickness, keratoconus, scleral thickness, swept-source optical coherence tomography

## Abstract

Purpose: To evaluate the relationship between choroidal thickness (CT) and anterior scleral thickness (AST) in patients with subclinical keratoconus (SKC) and established keratoconus (KC). Methods: This single-center prospective case-control study included 97 eyes of 97 patients: 44 KC eyes, 14 SKC eyes, and 39 age- and axial length (AL)-matched healthy eyes. Using swept-source optical coherence tomography, the AST was manually measured in four directions and the CT was obtained automatically from the Early Treatment Diabetic Retinopathy Study (ETDRS) grid. Principal component analysis (PCA) was used to linearly reduce the dimensionality of nine CT inputs to one significant component, CT1. A multivariate model was created to evaluate the association between CT1, AST, and several ocular parameters in SKC and KC patients. Partial correlation was then performed to adjust the confounding factors and to examine the effect of AST on CT1. Results: The PCA showed that CT1 accounts for 86.54% of the total variance in the nine original CTs of the ETDRS grid. The CT1 model was associated with age, AL, and AST in the superior meridian in SKC eyes, whereas in KC eyes, it was correlated with gender, age, AL, and AST in the inferior meridian (*p* < 0.001). The partial correlation between CT1 and AST in the superior zone was found to be significant, positive, and strong in SKC eyes (r = 0.79, *p* = 0.019), whereas a significant, positive, and moderate correlation between CT1 and AST at the inferior zone (r = 0.41, *p* = 0.017) was observed in KC eyes. Conclusions: Choroidal tissue was significantly correlated with the anterior sclera across the vertical meridian. This relationship was observed over the superior sclera in SKC eyes, whereas in established KC, it was over the inferior sclera. These results reveal new insights regarding the interactions between the anterior and posterior structures of the KC eyes and confirm the enigma of the pathophysiology of KC.

## 1. Introduction

Keratoconus (KC) is a progressive, bilateral, generally asymmetrical, and sporadic ectatic corneal disease characterized by paracentral corneal thinning and apical corneal protrusion [1]. KC induces irregular astigmatism and progressive myopia which causes an increase in higher-order aberrations (HOAs) and as a consequence, a decrease in visual acuity [1]. This visual impairment has a strong impact on the lives of many young patients, as it is commonly detected at a young age point, making this condition a significant public health problem [2]. Although the exact etiology is not still completely understood, it can be classified as an inflammatory corneal disorder [3], given the significant action of inflammatory mediators [4] and the increased oxidative stress [5,6]. Indeed, there are important risk factors related to inflammatory pathways (eye rubbing, allergy, atopy, and eczema) [7], as well as several systemic pathologies such as Down’s syndrome, Turner’s syndrome, and Leber’s congenital amaurosis [8] that enhance the occurrence and the progression of KC.

This underlying characteristic inflammatory mechanism of patients with KC [3,4,6] is evidence that this condition is not merely restricted to the corneal zone, but may affect other neighboring ocular tissues. Several studies have already described that alterations occur in the scleral geometric profile in KC patients, such as higher levels of scleral asymmetry [9] and steeping [10]. However, although there is evidence for the presence of deep changes in the scleral tissue and its loss of homogenization in corneal ectasia [11], the anterior scleral thickness (AST) in KC subjects seems to be similar to healthy subjects [12]. On the other hand, the implication of the internal ophthalmic structures in the pathogenesis of KC has been confirmed, which means that poor visual acuity might not be just attributed to corneal affection but also to retinal irregularities [13]. Despite some studies [14] observing similar central foveal thickness between KC and non-KC eyes, Özsaygılı et al. [15] reported that the inner plexiform layer was significantly thicker in KC than in non-KC eyes. Likewise, numerous investigations [14] have described a subfoveal choroidal thickening in KC eyes that goes beyond the fovea and affects several parafoveal sectors [16].

Hence, considering that the cornea and sclera are connective tissues [17] and the posterior sclera lies under the retina and the choroid [18], it can be assumed that the anatomical and morphological variations of these ocular structures extend from the anterior to the posterior segment of the eye, which might be partly explained by the chronic inflammatory events. To date, although some studies have investigated the association between corneal parameters versus scleral shape [19] and the correlation between corneal and scleral thickness [12], there is a lack of research about the association between anterior sclera and choroidal structure in KC subjects. Moreover, the coexistence of central serous chorioretinopathy (CSC) and KC has been described [20].

In this vein, the aim of the current study was to evaluate the relationship between AST and choroidal thickness (CT) in eyes with KC and SKC in order to determine if there is any connection between these two ocular tissues and to elucidate the interactions between the anterior and posterior structures of KC eyes.

## 2. Materials and Methods

This was a prospective case-controlled single study carried out at the Foundation of Medical Ophthalmology (FOM), Valencia, Spain. The study adhered to the tenets of the Declaration of Helsinki, and approval was obtained from the Institutional Review Board of the FOM (approval number PI093). All participants provided written informed consent prior to inclusion. All patients were recruited from the FOM and Optic, Optometry and Visual Sciences Department at the University of Valencia, Valencia, Spain.

Only data from one eye was included from each patient. The data from a single eye was randomly included if both eyes were positive for KC or SKC. Inclusion criteria were age between 18 and 55 years old, in order to avoid the impact of age on CT [21] and AST [22], and an axial length (AL) from 22 to 26 mm, considering the negative relation with CT [23] and AST [24]. The ocular assessment included corrected distance visual acuity (CDVA), refractive error, slit lamp biomicroscopy, intraocular pressure (IOP), AL (IOL Master 700; Carl Zeiss Meditec, Jena, Germany), tomographic analysis (Pentacam^®^ High Resolution, HR; Oculus Optikgeräte GmbH, Wetzlar, Germany) and fundus examination.

Patients who had mean spherical equivalent (SE) between −5 diopters (D) and +2 D, CDVA ≥ 0.0 log MAR (20/20), and no clinical signs on slit lamp examination, were included in the control group (CG). The diagnosis of SKC or KC was confirmed by a corneal specialist (CPM). The SKC group (SKG) was characterized by the absence of clinical (keratometric or biomicroscopic) signs of KC, a CDVA of 0.0 log MAR (20/20), and a fellow eye with established KC. The diagnosis of KC was made if there were at least one KC clinical sign on the slit lamp (Fleischer ring, Vogt striae, Munson Sign, stromal thinning, or anterior stromal scar) and an irregular cornea with an asymmetric bowtie pattern in the tomographic analysis. Patients with any history of ocular trauma, retinal or other eye diseases, previous crosslinking (CXL) procedure or other ocular surgery, and who had used contact lenses and topical medications at least two weeks before the examination, were excluded from the study. Patients with atopy or any dryness and inflammation corneal disease were also excluded because of the possible influence of the inflammatory cargo in their orbit and the correlation with CT [25].

The Belin–Ambrósio enhanced ectasia display (BAD-D index) and topographic KC classification (TKC) from Pentacam^®^ HR were used to stratify the sample. The Pentacam^®^ HR software (version 1.21r59) established a cut-off value of 1.6 and 3.0 to distinguish between healthy, SKC, and KC eyes. However, for expediency, we adjusted the cut-off value to 1.3 and 3.5 D to have balanced control and KC groups (KG). Hence, each eye was classified as follows: CG with BAD-D < 1.30 D; SKG with BAD-D ≥ 1.3 and <3.5 D and TKC = possible or nothing; and KG with BAD-D ≥ 3.5 and TKC ≥ 1. Corneal parameters provided by Pentacam^®^ HR, such as thinnest corneal thickness (TCT), maximum keratometry (Kmax), and total corneal astigmatism (ACt), were also included.

### Experimental Procedure: Scleral and Choroidal Imaging

The anterior and posterior exploration was performed after the initial screening once it was checked that all individuals fulfilled the inclusion and exclusion criteria. The scleral and choroidal analysis was conducted by a skilled optometrist (NBG) in a time-slot between 3 p.m. and 7 p.m. to avoid the diurnal variations of AST [26] and CT [27]. Scleral, choroidal imaging, and data analysis were extensively described in our previous works [12,16]. However, a brief summary is provided below.

The Casia 2 swept-source optical coherence tomography (Tomey, Nagoya, Japan) was used to measure the thickness of the nasal, temporal, superior, and inferior anterior sclera. The anterior global scan method was used to acquire all images which lie in a radial scan of 16 cross-sectional images averaged from 128 images with a scan resolution of 800 A-scan, a resolution image of 800 × 11,000 pixels per inch (PPI) per line sampling, and a depth of 6 mm. While patients fixated on a non-accommodative target placed 2 m away and about 20° out off-axis, the examiner paid close attention to measuring close to the limbus zone and to placing the horizontal (nasal and temporal sclera) or vertical (superior and inferior sclera) line from the radial scan aligned upon the sclero-corneal reflex. In order to distinguish all the structures, the B-scan image was modified to include the iris and to quantify the AST (Figure 1). The AST was manually quantified using a caliper once anatomical landmarks were identified on the B-scan (Figure 1A). The AST was defined as the axial distance between the inner boundary, identified as the hypo-reflective anterior wall of the ciliary body tissue, and the outer boundary, identified as the hypo-reflective conjunctival episcleral vessels. Three consecutive AST measurements were performed from the scleral spur every 1 mm (Figure 1B), but the mean of these measurements was calculated to perform the current analysis.

Choroidal images were performed using the DRI-1 swept source (Topcon Medical, Tokyo, Japan). The 12 radial-line scan pattern centered on the fovea was used to obtain the quantitative analysis of CT values. The device automatically identifies the outer boundary, the retinal pigment epithelium (RPE), the inner boundary, and the choroid-sclera junction, and provides the CT in a three-dimensional topographic map with nine subfields defined by an ETDRS-style grid (Figure 2). The subfoveal macula (1 mm central ring, cCT) involves the inner ring (3 mm diameter), which includes the nasal inner macula (iNAS), superior inner macula (iSUP), temporal inner macula (iTEM), and inferior inner macula (iINF). The outer ring (6 mm diameter) involves the nasal outer macula (oNAS), superior outer macula (oSUP), temporal outer macula (oTEM), and inferior outer macula (oINF). The quality of the choroidal images was registered with the choroidal index quality (CIQ) and only scans with CIQ ≥ 60 were included in the current study.

## 3. Statistical Analysis

Statistical data was assessed using SPSS software version 28.0 (Chicago, IL, USA) for Mac OS. Univariate analyses were used for descriptive purposes, using the mean and standard deviation (SD) for quantitative variables, and n (percentage) for categorical variables. Normality was verified using the Shapiro–Wilk test and graphical approaches. A comparison of demographic and basic clinical features between the three study groups was performed using one-way analysis of variance (ANOVA) or Kruskal–Wallis tests and Fisher exact test for categorical data. In order to reduce the nine CT subfields of the ETDRS grid included in the study, a principal component analysis (PCA) was used to create a new component, CT1, which accounted for most of the variance in the nine original CTs. The association between gender, age, AL, Kmax, TCT, CT1, and AST were assessed using multiple regression analysis in each group. Then, partial correlation was calculated for significantly relevant factors. The level of significance considered was set at 0.05 for all tests (two-tailed).

## 4. Results

The study included 97 eyes from 97 patients, 44 eyes from 44 KC patients, 14 eyes from 14 SKC patients, and 39 age- and AL-matched healthy subjects (KG 30.8 ± 8.4 years, SKG 32.6 ± 9.1 years, CG 30.1 ± 10.8 years, H(2) = 1.56, *p* = 0.45; KG 23.93 ± 1.31 mm, SKG 24.31 ± 0.89 mm, CG 23.6 ± 0.85 mm, F(2, 94) = 2.92, *p* = 0.06). The demographic and clinical characteristics by groups are shown in Table 1.

The mean AST of the three consecutive measurements every 1 mm across the nasal, temporal, superior, and inferior meridians were summarized in Table 2. One-way ANOVA revealed that there were no statistical differences between all three groups in any explored meridian (F(2, 81) = 1.80, *p* = 0.17).

Significantly choroidal thickening was found in the central ring (331 ± 56 μm vs. 300 ± 53 μm, *p* = 0.006) and in eight out of nine parafoveal sectors of the ETDRS grid between KG and CG (all macular areas, *p* < 0.05), except in the outer ring of the nasal subfield (*p* = 0.18). Although an increased CT profile in SKC eyes was observed (cCT: SKG 335 ± 56 μm vs. CG 300 ± 53 μm), these differences were not statistically significant (*p* = 0.45). The PCA results showed that one component (CT1), which accounted for accumulative 86.54% of the original nine CT subfield sets variability of the ETDRS grid, was extracted. Table 3 illustrates the component matrix and summarizes the coefficient weights extracted from each CT subfield.

The multivariate regression models for both groups, KG and SKG, are illustrated in Table 4 and Table 5, respectively. In KG, there was an association of CT1 with gender, age, AL, and AST over the inferior meridian (*p* < 0.05 in all variables), whereas SE, Kmax, and TCT were not linked with CT1 (*p* > 0.05). In SKG, the regression model revealed an association between CT1, age, AL, and AST over the superior region (*p* < 0.05 in all variables). Healthy patients did not show any association between CT1, demographic data, and ocular features (*p* > 0.05).

The partial correlation controlling gender, age, and SE showed a significant, strong, and negative relationship between CT1 and AL in SKC eyes (r = −0.70, *p* = 0.018) and KC eyes (r = −0.48, *p* = 0.01). The partial correlation adjusted for gender and AL demonstrated a negative and moderate association between CT1 and age in KC eyes (r = −0.30, *p* = 0.05). Although SKC eyes did not exhibit a significant link, a negative trend was observed (r = −0.54, *p* = 0.06). Figure 3 illustrates the partial scatterplot representing the interrelationship between CT1 and AL (Figure 3A) and age (Figure 3B) by main study groups corrected for the confounding factors.

The partial correlation adjusted for gender, age, AL, and SE between the new component CT1 and AST over the nasal, temporal, superior, and inferior scleral regions was plotted in Figure 4. There was a significant, positive, and strong association between CT1 and AST over the superior region in SKC eyes (r = 0.79, *p* = 0.019), whereas KC eyes revealed a significant, positive, and moderate association between CT1 and AST over the inferior scleral region (r = 0.41, *p* = 0.017). Healthy patients showed a negative trend regarding the relationship between CT1 and AST over the temporal meridian (r = −0.38, *p* = 0.05). There was no significant correlation between CT1 and AST over nasal meridian in any study group.

## 5. Discussion

The findings of the present study demonstrated that there was a significant, positive, and moderate correlation in KC eyes over the inferior sclera, whereas a significant, positive, and strong correlation was observed over the superior sclera in SKC eyes when confounding factors such as gender, age, SE and AL were controlled. Instead, healthy eyes showed a negative correlation trend over the nasal scleral zone.

It has been broadly described that KC eyes suffer a subfoveal choroidal thickening [13,14,15,28]. In fact, we recently demonstrated that CT was increased under the fovea and beyond [16], which confirmed the role of the choroidal structure in the pathophysiology of KC disease. In this study, we examined the factors associated with choroidal structure in SKC and KC patients, specifically with scleral thickness. As a result, we observed that those patients whose choroids were thickened due to the presence of the pathology seemed to have thicker anterior sclera in the lower zone; meanwhile, when the pathology was not yet well-established, this direct relationship could be found in the upper part of the anterior sclera. In contrast, healthy patients with thicker choroids presented thinner AST in the temporal scleral zone. To the best of our knowledge, no previous studies have described these results regarding the relationship between these two structures in KC disease.

In the current study, KC and SKC eyes did not show scleral thinning in any meridian explored compared to control healthy patients, as we reported in our preliminary investigations [12]. Considering that peripheral corneal thinning was observed outside the ectatic region toward the limbus equator [29], the alterations described in the scleral tissue of KC patients, such as homogeneity loss in collagen fibers [11] and increased corneo-scleral shape [9,10], and the similar collagen composition of both connective tissues, sclera, and cornea, we could hypothesize that a scleral thinning occurs. However, in light of these results, the anterior sclera did not take part in the natural evolution of the disease, as Schlatter et al. [30] also described.

The results of the multivariate regression models revealed that AL and age were factors associated with CT1 in both KG and SKG groups. There was a negative relation between AL, age, and CT1, which is consistent with the results of previous studies in healthy [21,23] and KC patients [28], where it was demonstrated that greater AL and age yield to a thinner choroid. Males with SKC eyes presented thinner CT than females, in contrast to KC patients, where gender was not linked with CT1. This could be attributed to the unbalanced ratio of males/females of SKC (4/10) compared to KC eyes (22/22), the small sample size of SKC patients (14 SKC versus 44 KC eyes), and the significant gender difference between the three groups of patients. The anterior sclera was positively linked with AST in pathological eyes at different scleral areas. The model with SKC eyes revealed that there was a link between CT1 and the superior AST; meanwhile, the model created with KC showed that this link was with the inferior AST. It has to be noted that the association was over the vertical meridian, where the scleral thickness was thinnest (superior) and thickest (inferior) in comparison to nasal and temporal thicknesses. Moreover, we could attribute these differences to the inferior-superior scleral thickness asymmetry, which was statistically different between KC and control, specifically between SKC eyes and healthy patients, as the asymmetrical thickness was greater in KC eyes than in SKC eyes [12]. Consistent with Gutierrez-Bonet et al. [28], K-max and TCT were not associated with CT1. Likewise, SE did not show any correlation with CT1. Flores-Moreno et al. [23] described that SE only correlated in highly myopic eyes whose SE ranged between −6 and −24 D, which were not included in our study.

In order to obtain the correlation between the anterior sclera and choroid without confounding factors, we calculated the relationship controlling gender, age, and AL. Moreover, despite the fact that SE was not associated with CT1, we also considered this factor because AST was correlated with SE in healthy [31] and KC eyes [12]. The findings showed that the order and the scleral zone of the partial correlation were different depending on whether we observed healthy, preclinical KC, or established KC eyes. The correlation trend observed in healthy patients at the temporal sclera could be explained by our prior results, where we found variations of scleral thickness with scleral eccentricity only in this zone in comparison to the nasal, superior, and inferior scleral sectors [12]. The adjusted correlation in SKC was stronger than in KC eyes, but this could be attributed to the number of eyes included in the SKG compared to the KG. Indeed, KG gathered eyes with different KC degrees, and the relationship could be affected by this issue. Further investigations including a greater number of SKC patients and patients with the same KC degree are necessary to confirm this difference. On the other hand, it should also be considered that the order of the correlation was greater because the order of the adjusted correlation between CT1 and AL was stronger in SKG (r = −0.70, *p* = 0.006) than in KG (r = −0.49, *p* < 0.001). Moreover, the cone is normally localized over the infero-temporal cornea [1]. Hence, in light of the current results, a scleral thickening in the same area could be taking place when the choroid is thickened. These findings are very interesting considering that in the disease process, the cornea becomes thinner. However, scleral tissue seems to thicken, as happens with the thickening of the choroidal structure. This could be attributed to collagen reorganization and homogenization loss [11] due to tension because of the forces being taken on the eye surface as a compensating mechanism. The role of the inflammation on the eye surface of KC currently remains an enigma that needs further immunohistological investigations.

One of the proposed theories that explains choroidal thickening in KC was the inflammatory events of the disease process. An increase has been found in inflammatory cells in the choroidal stromal which causes stromal infiltration and vascular dilatation, with the latter being the main cause [13,14,16,28]. This choroidal vascular dilatation was the main cause of the choroidal thickening in other pathologies such as CSC. Interestingly, Fernández-Vigo et al. [32] described that those patients affected by CSC presented an increase in the scleral thickness at the most anterior portion in comparison to the healthy population. Similarly, Imanaga et al. [33] described that this scleral thickening of CSC subjects also occurred in the intermediate portion of the sclera. Most recently, the same authors [34] described that CT and scleral thickness were significantly and positively correlated with the luminal/stromal ratio of the choroidal structure which, in part, was in concordance with the findings of the current study in SKC and KC subjects. Moreover, it should be taken into account that there was evidence about the coexistence of CSC and KC [20]. However, due to the fact no differences have been found between AST of non-KC and KC patients, further investigations are needed to confirm these assumptions.

This study has some limitations. Healthy, SKC, and KC subjects were not gender-matched; however, KC disease did not distinguish between male and female [1,8], and there is controversy in the literature about gender influence in AST and CT [22,27,30]. Since the scleral tissue surface was not parallel, Laplace, refractive indices, and B-Scan tilt corrections should be applied to minimize the optical distortion [35]. Another aspect to consider is that most of the patients were Caucasic, which may also limit the generalization of these results to other populations, considering the close relationship between KC and lineage [36]. The bounded sample size of SKC and KC could be another limitation, as well as the unbalanced size of KC subgroups, but it should be taken into consideration that KC is a rare disease with lower prevalence in Caucasian patients.

## 6. Conclusions

Choroidal tissue was significantly correlated with the anterior scleral thickness at the vertical meridian. This relationship was observed over the superior sclera in SKC eyes, whereas in established KC, it was observed over the inferior sclera. Those patients whose choroid was thickened due to the presence of the pathology seem to have thicker anterior sclera. These results reveal new insights regarding the interactions between the anterior and posterior structures of the KC eyes, confirming the enigma of the pathophysiology of KC and the possible role of scleral thickness in KC pathogenesis.

## Figures and Tables

**Figure 1 diagnostics-14-02280-f001:**
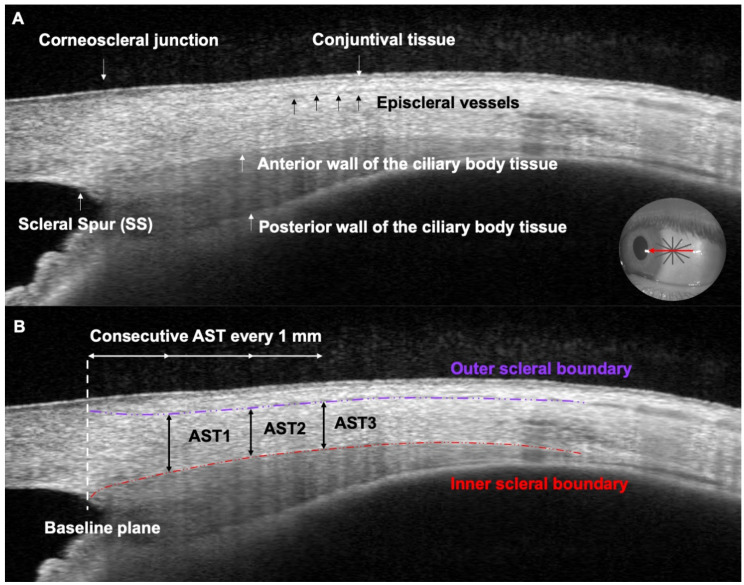
Sequential image analysis of the anterior scleral thickness (AST) measurement process. (**A**) B-Scan of the anterior sclera where all the anatomical structures are marked. The circle image (right-bottom) illustrates the en-face image with the radial scan over the temporal sclera where the red arrow represents the single line cutting across the sclero-corneal reflex selected from the 16 cross-sectional B-scans. (**B**) Anterior scleral image where three consecutive measurements of AST every 1 mm were performed from the baseline plane, defined from the scleral spur point. The AST was manually quantified as the axial distance between the inner boundary (purple dashed line), identified as the hypo-reflective conjunctival episcleral vessels, and the outer boundary (red dashed line), identified as hypo-reflective anterior wall of the ciliary body tissue.

**Figure 2 diagnostics-14-02280-f002:**
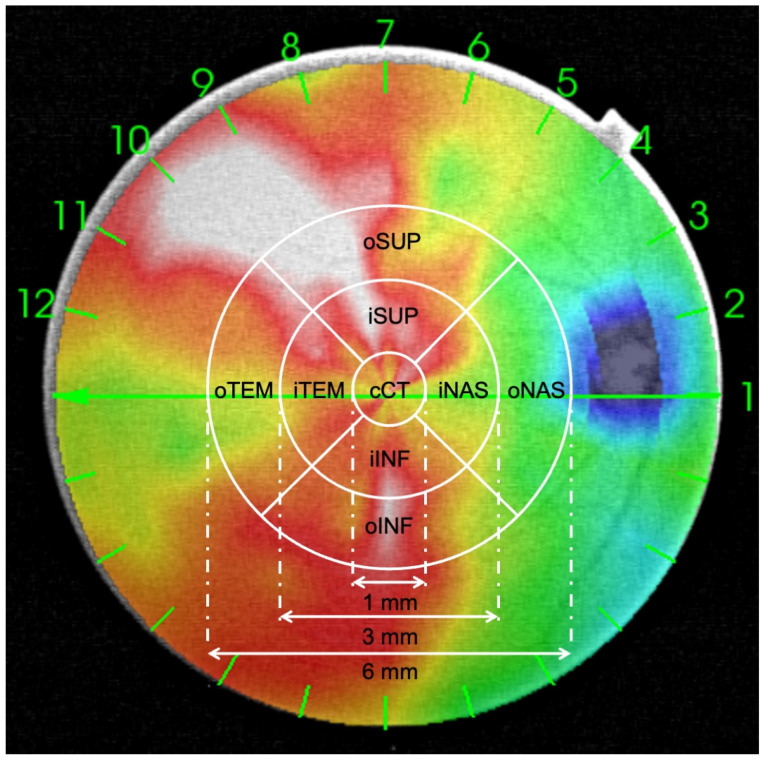
Choroidal thickness (CT) map in a grade 2 keratoconus eye. CT was automatically provided by the device in the central ring (1 mm in diameter), in the inner ring (3 mm in diameter), and in the outer ring (6 mm in diameter) across several macular sectors defined by the Early Treatment Diabetic Retinopathy Study (ETDRS). The central ring includes the central choroidal thickness (cCT). The inner ring includes the inner nasal macula (iNAS), inner temporal macula (iTEM), inner superior macula (iSUP), and inner inferior macula (iINF). The outer ring includes the outer nasal macula (oNAS), outer temporal macula (oTEM), outer superior macula (oSUP), and outer inferior macula (oINF).

**Figure 3 diagnostics-14-02280-f003:**
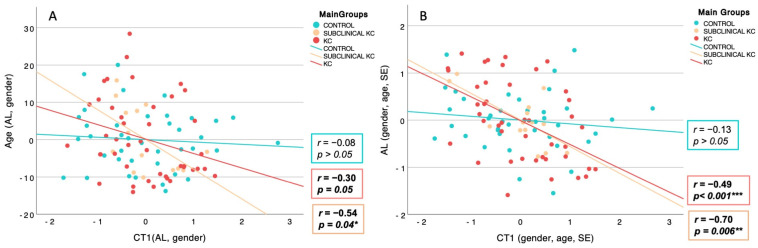
Partial scatterplot representing the linear relationship of the main groups between the new component of choroidal thickness (CT1) and: (**A**) age-adjusted for gender and axial length (AL); (**B**) axial length adjusting for gender, age, and spherical equivalent (SE). The right inferior boxes illustrate the partial correlation of control (blue), subclinical keratoconus (yellow), and keratoconus (red). Significant correlations are indicated by an asterisk (*), * *p*-value < 0.05, ** *p*-value < 0.01, *** *p*-value < 0.001.

**Figure 4 diagnostics-14-02280-f004:**
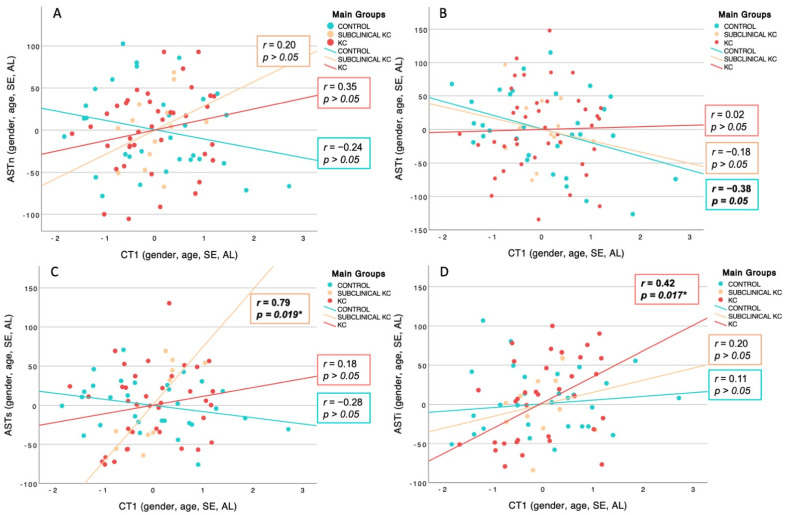
Partial scatterplot adjusted for age, gender, axial length (AL), and spherical equivalent (SE) which represents the linear relationship within the three main groups, as far as the new component of choroidal thickness (CT1) and anterior scleral thickness (AST) across the four meridians explored are concerned: (**A**) nasal, (**B**) temporal, (**C**) superior and (**D**) inferior. The right inferior boxes illustrate the partial correlation of control (blue), subclinical keratoconus (yellow), and keratoconus (red). Significant correlations are indicated by an asterisk (*), * *p*-value < 0.05.

**Table 1 diagnostics-14-02280-t001:** Demographic and clinical features of each group, healthy, subclinical, and keratoconus eyes. Quantitative variables are described as mean value ± standard deviation (range), and categorical variables as absolute frequencies (*n*) and percentage (%).

Parameters	CG (*n* = 39)	SKG (*n* = 14)	KG (*n* = 44)	*p*-Value
Gender (female)	10 (25.6%)	10 (71.4%)	22 (50%)	0.006 ** ^
Race: White	39 (100%)	14 (100%)	37 (84.1%)	0.17 ^
Arab people	-	-	4 (9.1%)	
People of African	-	-	2 (4.5%)	
descent				
South Asian	-	-	1(2.3%)	
Age (years)	30.8 ± 8.4	32.6 ± 9.1	30.1 ± 10.8	0.45 ^‡^
	(18 to 55)	(18 to 55)	(18 to 55)	
SE (D)	−0.74 ± 1.2	−1.73 ± 2	−2.32 ± 2.75	<0.001 *** ^‡^
	(−4.63 to 0.75)	(−5.63 to 0.63)	(−13.9 to 1.0)	
CDVA, logMAR	−0.07 ± 0.07	−0.02 ± 0.08	0.11 ± 0.20	<0.001 *** ^‡^
	(−0.20 to 0)	(−0.20 to 0)	(−0.10 to 0.82)	
KC severity:	-	-		-
Grade 1			18 (40.9%)	
Grade 2			19 (43.2%)	
Grade 3			7 (15.9%)	
AL (mm)	23.60 ± 0.85	24.31 ± 0.89	23.93 ± 1.31	0.06 ^†^
	(22.21 to 25.76)	(22.60 to 25.76)	(22 to 25.90)	
K max (D)	43.3 ± 1.59	43.8 ± 1.59	50.18 ± 4.32	<0.001 *** ^†^
	(40.4 to 46.7)	(41.4 to 47.6)	(43.2 to 59.6)	
ACt (D)	0.73 ± 0.50	0.71 ± 0.54	2.93 ± 1.68	<0.001 *** ^‡^
	(0.1 to 2.3)	(0.1 to 1.8)	(0.1 to 9.3)	
TCT (μm)	541 ± 34	516 ± 33	469 ± 35	<0.001 *** ^†^
	(484 to 618)	(464 to 583)	(403 to 540)	
BAD-D (mm)	0.68 ± 0.59	2.07 ± 0.58	6.62 ± 3	<0.001 ** ^‡^
	(−0.54 to 1.82)	(1.3 to 3.8)	(1.52 to 14.43)	

^(†)^ one-way analysis of variance (ANOVA); ^(‡)^ Kruskal-Wallis test; ^(^)^ Chi square test, ** *p*-value < 0.01, *** *p*-value < 0.001. CG: control group, SKG: subclinical keratoconus group, KG: keratoconus group, SE: spheric equivalent, D: diopters, CDVA: corrected distance visual acuity, KC: Keratoconus, AL: axial length, K max: maximum keratometry, ACt: total corneal astigmatism, TCT: thinnest corneal thickness, BAD-D: Belin–Ambrosio index.

**Table 2 diagnostics-14-02280-t002:** Mean anterior scleral thickness of the three consecutive scleral measurements every 1 mm over the four explored scleral zones: nasal, temporal, superior, and inferior.

AST	CG (*n* = 39)	SKG (*n* = 14)	KG (*n* = 44)	*p*-Value
Nasal	534 ± 53	510 ± 51	532 ± 532	0.17 ^†^
	(449 to 628)	(420 to 579)	(390 to 648)	
Temporal	509 ± 65	509 ± 63	536 ± 63	0.23 ^†^
	(378 to 670)	(431 to 633)	(404 to 697)	
Superior	446 ± 34	477 ± 51	463 ± 50	0.12 ^†^
	(378 to 503)	(407 to 572)	(371 to 592)	
Inferior	572 ± 47	568 ± 52	575 ± 68	0.92 ^†^
	(447 to 686)	(476 to 641)	(431 to 756)	

^(†)^ One-way ANOVA: analysis of variance; AST: anterior scleral thickness, CG: control group, SKG: subclinical keratoconus group, KG: keratoconus group.

**Table 3 diagnostics-14-02280-t003:** Results of the principal component analysis (PCA) for the choroidal thickness of the nine subfields of the ETDRS grid. CT1 represents the new component that explains a cumulative 86.54% of the total amount of variance.

Matrix Component	Principal Component (CT1)	Total of Variance Explained Extraction Sum of Squared Loadings	
		Variance %	Cumulative %
cCT	0.986	86.54	86.54
iNAS	0.952		
iTEM	0.942		
iSUP	0.934		
iINF	0.809		
oNAS	0.879		
oTEM	0.859		
oSUP	0.859		
oINF	0.873		

CT1: Choroidal thickness 1; ETDRS: Early Treatment Diabetic Retinopathy Study; cCT: central choroidal thickness, iNAS: inner nasal macula, iTEM: inner temporal macula, iSUP: inner superior macula, iINF: inner inferior macula, oNAS: outer nasal macula, oTEM: outer temporal macula, oSUP: outer superior macula, oINF: outer inferior macula.

**Table 4 diagnostics-14-02280-t004:** Results of the stepwise multiple linear regression model in subclinical keratoconus eyes between the new component CT1, anterior scleral thickness (AST) in the four scleral zones, and several ocular parameters. The model summary showed an adjusted R^2^ of 86.2%.

Model Summary	R	R^2^	Adjusted R^2^	Error	Durbin–Watson	
CT1	0.969	0.937	0.862	0.53	2.105	
**Independent variables**	**B coefficient** **(Std error)**		**95% Confidence Interval**		***p*-value**	**Importance**
			**Lower**	**Upper**		
Intercept	23.95 (3.70)		15.41	32.48	0.001 ***	
AL	−1.09 (0.15)		−1.44	−0.74	0.001 ***	0.70
ASTs	0.009 (0.003)		0.003	0.014	0.009 **	0.16
Age	−0.05 (0.02)		−0.09	−0.01	0.013 *	0.14

*** *p*-value < 0.001. ** *p*-value < 0.01. * *p*-value < 0.05. CT1: choroidal thickness 1, AL: axial length ASTs: anterior scleral thickness superior, Std: standardized.

**Table 5 diagnostics-14-02280-t005:** Results of the stepwise multiple linear regression model in keratoconus eyes between the new component CT1, anterior scleral thickness (AST) at the four scleral zones, and several ocular parameters. The model summary showed an adjusted R^2^ of 43.2%.

Model Summary		R	R^2^	Adjusted R^2^	Error	Durbin–Watson	
CT1		0.63	0.45	0.432	0.84	1.92	
**Independent variables**		**B coefficient** **(Std error)**		**95% Confidence Interval**		***p*-value**	**Importance**
				**Lower**	**Upper**		
Intercept		9.90 (2.81)		4.19	15.61	0.001 **	
AL		−0.48 (0.11)		−0.71	−0.24	<0.001 ***	0.47
Age		−0.04 (0.01)		−0.07	−0.01	0.005 **	0.24
ASTi		0.005 (0.002)		0.001	0.01	0.014 *	0.18
Gender	Men	−0.54 (0.27)		−1.08	−0.003	0.049 *	0.11
	Women	^†^					

^†^ Reference value to compare between categorical variables. *** *p*-value < 0.001. ** *p*-value < 0.01. * *p*-value < 0.05. CT1: choroidal thickness new component, AL: axial length ASTi: anterior scleral thickness inferior meridian, Std: standardized.

## Data Availability

Data are available on request from the corresponding author, Neus Burguera-Giménez (neus.burguera@uv.es).

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
