# Peer review of "Relationship between Choroidal Thickness and Anterior Scleral Thickness in Patients with Keratoconus"

_diagnostics, 2024, doi:10.3390/diagnostics14202280_

Round 1

Reviewer 1 Report

Comments and Suggestions for Authors

Dear authors,

I would like to thank you for this very interesting and novel studies which could reveal new aspects in the pathophysiology of KC. Here are my comments on your manuscript

A check from an English professional is advised. There are some points which need careful assessment and correction like the following:

Abstract (lines 25-30) check gaps eg 85.7% or 85.7 %, r=0.79 or r= 0.79 etc,

check gaps eg line 155 1mm or 1 mm and accordingly throughout the text, Line 220 ..

Line 28 The partial correlation between CT1 and AST in the superior zone (..) was found to be significant, positive and strong, whereas a positive and moderate correlation between CT1 and AST at the inferior zone (…) was observed in KC eyes.

Check gaps before reference number throughout the text (eg line 44 health problem [2], not health problem[2], line 52 etc..

Line 43, at whom (not when)

Line 44 a significant public health problem (not real)

Line 47 and the increased oxidative stress..

Line 48 ..as well as… (remove , before and after phrase)

Line 50 instead: that enhance the occurrence and progression of KC

Line 55 remove …

Line 61 double gap between [14] and observed

Line 62 and 63 non-KC eyes…

Line 73 Moreover, the coexistence (..) has been described

Line 75 was instead of is

Line 76 and SKC in order to comprehend if there is any correlation between these two ocular tissues and in order to elucidate about the…

Line 84 prior to inclusion (remove the)

Line 89 ..was included from each patient.

Line 88 in order to avoid..

Line 92 (Pentacam® High Resolution, HR..)

Line 95, no clinical signs on slit lamp examination, were included…

Line 96.. by a corneal specialist (CPM).

Line 102 previous treatment with corneal crosslinking (CXL) procedure or other ocular surgery..

Line 104 atopy

Line 107 Belin-Ambrósio..

Line 108, 113 Pentacam® HR

Line 110 we adjusted

Line 112 TKC = possible or nothing what do you mean by that? When no value is screened?

Line 113 such as

Line 118 fullfilled

Line 129 measured carefully with the highest possible accuracy..

Line 143 en face

Line 172 Statistical data was assessed

Line 178 In order to reduce

Line 181 were assessed

Line 183 The level of significance was set at 0.05

Table 1 Gender (female)

Race Middle East (instead of Arab?)

Line 200 stastistically significant choroidal thickening

Line 205 accounted

Line 206 was extracted, line 207 the coefficient..

Line 236 adjusted

Line 243 significant, positive and strong association..

Line 252 linear, within the three main groups, as far as the new component….are concerned.

Line 258, 259-260 significant, positive and moderate correlation…

Line 265 the role of the choroidal structure (or choroid) in the pathophysiology..

Line 266 we examined..267 we observed .. Line 270 could be found..

Line 288 males…females..

Line 306 we calculated (in general refer using simple past), check the whole text, avoid present perfect

Line 307 Moreover, despice the fact that SE was not associated…we also considered this factor because…

Line 318 it should be also considered

It is very interesting that although the cornea becomes thinner at KC, the sklera thickens, this could be attributed to collagen reorganization due to tension because of the forces being taken on the eye surface as a compensating mechanism. The role of the inflammation on the eye surface, even on the tear film which has been already investigated in KC compared to healthy corneas, is unknown. These parts need further (immunohistological) investigation. Maybe further comment in your manuscript.

Line 328 Vigo et al. [32] described…

Line 365 dot at the end of the phrase.

References

:1 Age-specific incidence and prevalence of keratoconus: A nationwide registration study (remove capitals) …169-172 (complete numbers) (accordingly for all references), reference 2, 7, 12, 13, 15, 16, 20, 23, 25, 27, 33, 34

2: Keratoconus: An inflammatory disorder? …843-859.

5 and for all references: remove month after year of publication

10, 19: Cont Lens Anterior Eye

25: Clin Ophthalmol. 2021;15:1799-1807.

Comments on the Quality of English Language

Dear authors,

I would like to thank you for this very interesting and novel studies which could reveal new aspects in the pathophysiology of KC. Here are my comments on your manuscript

A check from an English professional is advised. There are some points which need careful assessment and correction like the following:

Abstract (lines 25-30) check gaps eg 85.7% or 85.7 %, r=0.79 or r= 0.79 etc,

check gaps eg line 155 1mm or 1 mm and accordingly throughout the text, Line 220 ..

Line 28 The partial correlation between CT1 and AST in the superior zone (..) was found to be significant, positive and strong, whereas a positive and moderate correlation between CT1 and AST at the inferior zone (…) was observed in KC eyes.

Check gaps before reference number throughout the text (eg line 44 health problem [2], not health problem[2], line 52 etc..

Line 43, at whom (not when)

Line 44 a significant public health problem (not real)

Line 47 and the increased oxidative stress..

Line 48 ..as well as… (remove , before and after phrase)

Line 50 instead: that enhance the occurrence and progression of KC

Line 55 remove …

Line 61 double gap between [14] and observed

Line 62 and 63 non-KC eyes…

Line 73 Moreover, the coexistence (..) has been described

Line 75 was instead of is

Line 76 and SKC in order to comprehend if there is any correlation between these two ocular tissues and in order to elucidate about the…

Line 84 prior to inclusion (remove the)

Line 89 ..was included from each patient.

Line 88 in order to avoid..

Line 92 (Pentacam® High Resolution, HR..)

Line 95, no clinical signs on slit lamp examination, were included…

Line 96.. by a corneal specialist (CPM).

Line 102 previous treatment with corneal crosslinking (CXL) procedure or other ocular surgery..

Line 104 atopy

Line 107 Belin-Ambrósio..

Line 108, 113 Pentacam® HR

Line 110 we adjusted

Line 112 TKC = possible or nothing what do you mean by that? When no value is screened?

Line 113 such as

Line 118 fullfilled

Line 129 measured carefully with the highest possible accuracy..

Line 143 en face

Line 172 Statistical data was assessed

Line 178 In order to reduce

Line 181 were assessed

Line 183 The level of significance was set at 0.05

Table 1 Gender (female)

Race Middle East (instead of Arab?)

Line 200 stastistically significant choroidal thickening

Line 205 accounted

Line 206 was extracted, line 207 the coefficient..

Line 236 adjusted

Line 243 significant, positive and strong association..

Line 252 linear, within the three main groups, as far as the new component….are concerned.

Line 258, 259-260 significant, positive and moderate correlation…

Line 265 the role of the choroidal structure (or choroid) in the pathophysiology..

Line 266 we examined..267 we observed .. Line 270 could be found..

Line 288 males…females..

Line 306 we calculated (in general refer using simple past), check the whole text, avoid present perfect

Line 307 Moreover, despice the fact that SE was not associated…we also considered this factor because…

Line 318 it should be also considered

It is very interesting that although the cornea becomes thinner at KC, the sklera thickens, this could be attributed to collagen reorganization due to tension because of the forces being taken on the eye surface as a compensating mechanism. The role of the inflammation on the eye surface, even on the tear film which has been already investigated in KC compared to healthy corneas, is unknown. These parts need further (immunohistological) investigation. Maybe further comment in your manuscript.

Line 328 Vigo et al. [32] described…

Line 365 dot at the end of the phrase.

References

:1 Age-specific incidence and prevalence of keratoconus: A nationwide registration study (remove capitals) …169-172 (complete numbers) (accordingly for all references), reference 2, 7, 12, 13, 15, 16, 20, 23, 25, 27, 33, 34

2: Keratoconus: An inflammatory disorder? …843-859.

5 and for all references: remove month after year of publication

10, 19: Cont Lens Anterior Eye

25: Clin Ophthalmol. 2021;15:1799-1807.

Reviewer 2 Report

Comments and Suggestions for Authors

The study focused on the relationship between choroidal thickness (CT) and anterior scle-ral thickness (AST) in patients with subclinical keratoconus (SKC) and established keratoconus (KC). The results revealed news insights regarding the interactions between the anterior and posterior structures of the KC eyes and confirm the enigma of the pathophysiology of KC. It is an interesting study. However, some points are to be addressed before it is accepted.

 1 There are three groups, how to identify each group? For example, how to distinguish between subclinical keratoconus (SKC) and established keratoconus patientsIs it distinguished only by slit-lamp detection?

2 The experimental sample was too small, especially in the SKC group, with only 14 patients, which easily led to inaccurate statistical results.

3  Line 202-204, Significantly choroidal thickening was found in the central ring (331 ±56 vs. 300± 53 , p = 0.006) and in eight out nine parafoveal sectors of the ETDRS grid between KG and CG (all macular areas, p <0.05), except in the outer ring of the nasal subfield (p =0.18). Although an increased CT profile in SKC eyes was observed (cCT: SKG 335 ± 56 vs.  CG 300 ± 53 ), these differences were not statistically significant (p=0.45).  Why not make a table to explain it? 

Comments on the Quality of English Language

Quality of English Language is good. Minor editing of English language required.
